# The Effect of Different Storage Temperatures over Time on the pH of Mammary Gland Secretions in Periparturient Mares

**DOI:** 10.3390/ani14172598

**Published:** 2024-09-06

**Authors:** Igor F. Canisso, Gabrielle B. A. G. Amorim, Humberto B. Magalhaes

**Affiliations:** 1Department of Veterinary Clinical Medicine, College of Veterinary Medicine, University of Illinois Urbana-Champaign, Urbana, IL 61802, USA; 2Department of Veterinary Surgery and Animal Reproduction, School of Veterinary Medicine and Animal Science, São Paulo State University, Botucatu 01049-010, Brazil

**Keywords:** foaling, impending parturition, equine, horse

## Abstract

**Simple Summary:**

Mammary gland secretions (MGS), electrolytes, and pH are used to detect impending parturition in mares. Questions remain about whether all mares present sodium potassium inversion at labor. This study demonstrated that all mares had sodium potassium inversion and acidic pH at the time of parturition. Milk pH is neutral in the first seven days postpartum; the clinical significance of this remains to be determined. The pH of MGS can be measured with minimal variation stored at three different temperatures, except when the pH is ~7.5, which went up to ~8 immediately after the storage; however, the clinical significance of this variation can likely be negligible. This present study is the first to address these two physiological and practical questions about the pH of MGS in periparturient mares.

**Abstract:**

The objectives of this study were (i) to determine pH and electrolyte concentrations in MGS collected prepartum and at parturition, (ii) to characterize mare milk pH during the first week postpartum, and (iii) to evaluate pre-foaling MGS pH at three storage temperatures. This study outlined two hypotheses: (i) all mares exhibit acidic pH, increased calcium, magnesium, and potassium, and reduced sodium concentrations regardless of prepartum pH and electrolytes; (ii) pre-foaling MGS pH varies with storage temperature and time in an initial value-dependent manner. Twenty-three multiparous mares were monitored daily from 320 days of gestation until parturition. Pre-foaling MGS was collected, and pH was immediately measured using a hand-held pH meter. Aliquots were preserved for further electrolyte analysis. Postpartum, samples from day −7 to 0 (day of foaling) were thawed, and electrolyte concentrations (calcium, magnesium, sodium, potassium) were determined. For the three storage temperatures, pH was measured at 0, 15, 30, 45, and 60 min after storage, and hourly for 10 h post-collection. A range of pH 8 to 6.5 was included to avoid bias towards a specific pH value. The chosen pH groups were 8 (range 7.8–8.2), 7.5 (range 7.3–7.7), 7 (6.7–7.2), and 6.5 (6.2–6.6). Overall, storage temperature affects pH (*p* < 0.05). In conclusion, this study demonstrated that the majority of the mares had sodium–potassium inversion and acidic pH at foaling. Milk pH is neutral up to four days after foaling, becoming slightly alkaline afterwards, with undetermined clinical significance. The pH of MGS showed minimal variation across storage temperatures, except for pH ~7.5, which increased to ~8 post-storage. This study is the first to address these physiological and practical questions about MGS pH in periparturient mares.

## 1. Introduction

While most equine parturitions are uneventful, approximately 10% of foaling mares experience dystocia [1,2], and attending foaling is paramount to cope should dystocia occur. Fortunately, most of the equine dystocia may be resolved on-farm if qualified foaling assistance is available during parturition; while large broodmare farms have the luxury of constant monitoring by trained staff, such aid may not be feasible for small farms foaling one or a few mares [3,4]; thus, methods to detect the day of parturition are necessary for the latter type of farms.

As the duration of pregnancy is highly variable (e.g., 310 to ~400 days) in mares delivering normal foals, and physical body changes associated with parturition are unreliable, monitoring of the periparturient mare may be carried out for an extended period before parturition [3,4,5]. Most equine parturitions occur at night [6] when personnel are reduced on most farms [4]. In the last 40 years, several tools have been developed to aid in foaling monitoring. Positional monitoring devices (e.g., Birthalarm©, Gallagher Europe B.V., Groningen, NL, USA, EquiPage^©^, Kee-port, Inc., Arlington, WA, UAS, Breeder Alert^©^, Allsman Enterprises, LLC, Grants Pass, OR, USA) and a vulva apparatus (Foal-Alert^©^, Foalert Inc., Acworth, GA, USA) have been incorporated into the routine monitoring of pre-foaling mares [4,7]. Positional devices assume that mares will not stay in lateral recumbence for more than 2.5–3 min unless foaling; if so, the mare is deemed to be in active foaling, and an alarm is activated [7]. Foal-Alert^©^ is based on the principle that the vulvar lips will be separated during active labor, thus pulling a transponder off a magnet and activating a transmitter [7]. While these devices may be helpful for some farms, multiple false alarms may exist; for example, the alarm may fail if the mare foals standing (positional monitoring devices) or fetal mal-disposition (e.g., a breech foal) prevents the vulva from separating and displacing the magnet from the transmitter of the Foal-Alert device. In many foaling programs, different systems may monitor mares to ensure assistance can be provided during parturition.

Serial assessment of mammary gland secretion (MGS) has been used to predict parturition in mares carrying normal pregnancies. Several studies out of the UK [8,9] demonstrated that mares close to parturition have a decrease in sodium and an increase in potassium, calcium, and magnesium; these findings have been confirmed by more recent studies out of New Zealand and Kentucky [3,10]. Since measuring all electrolytes is expensive and results may not be returned promptly, mare-side tests have been developed [11,12]. Water hardness (an indirect measure of calcium and magnesium) strip tests have been described [3,8,11]. Yet, this type of test is not commonly used in practice [4]. Calcium carbonate (e.g., Foal-Watch) content is a traditional method used worldwide to determine the likelihood of a mare foaling [12]. However, these horse-side calcium tests are expensive and require at least 1–3 mL of MGS, which some mares may not have until the time of parturition [3,4]. 

Colostrum is the first and foremost source of nutrition, immunoglobulins, and biological compounds for a newborn foal; thus, a good quality colostrum is paramount to protect the foal against pathogens in the first weeks of life. The breed of the mare, health status, age, parity, and feeding seem to influence the colostrum/milk composition [13]. Serial assessment of pH from the MGS has been used as a low-cost and reliable method to detect impending parturition in mares and, lately, in jennies [3,14,15,16]. Most mares carrying normal pregnancies foal with an acidic pH of 6.2–6.6 [3,4,14,16]. While the mechanism involved in reducing pH from MGS in mares and jennies remains to be determined [3,4,14,16], in cows, the reduction is mediated by increased expression of carbon anhydrase in mammary gland tissues [17]. The role of MGS pH in the periparturient mare and newborn foal is unclear. Still, it has been suggested that acidic pH may be necessary for the optimal passive transfer of immunoglobulins to neonates [14]. Anecdotally, it has been thought that milk pH remains acidic during the first week of life to facilitate the foal’s milk digestion. Still, it is unknown if mare colostrum and milk pH remain acidic during the first week of life. 

After the different studies assessing the pH of pre-foaling mares [3,14,16], a critical field question remains unanswered, “how long after milking a mare can one reliably measure pH from MGS secretions?” In all the studies, pH from MGS was determined within a few minutes of collection. Intuitively, it seems that the pH of MGS could vary remarkably several minutes after the collection due to changes in temperature, increase in bacteria, oxygenation, etc. Under field situations, assessing MGS’ pH minutes to hours after sampling would be helpful, allowing the practitioner or breeder to collect secretions and transport them to a location where a pH meter or test strips are available. Understanding the stability of MGS pH under different storage conditions is crucial for accurately predicting foaling under various field conditions.

The objectives of this study were (i) to determine pH and electrolyte concentrations from MGS collected prepartum and at the time of parturition, (ii) to characterize mare milk pH in the first week postpartum, and (iii) to evaluate pre-foaling MGS pH at three storage temperatures. Two hypotheses were outlined in this present study: (i) regardless of prepartum pH and electrolyte concentrations, at the time of parturition, all mares present acidic pH; increased concentrations of calcium, magnesium, and potassium; and reduced concentrations of sodium; (ii) varying pre-foaling mammary secretion pH varies with temperature of storage and time in an initial value-dependent manner. 

## 2. Materials and Methods

### 2.1. Animals

Twenty-three light-breed multiparous broodmares presented for foaling management to the Equine Theriogenology Service, Veterinary Teaching Hospital, College of Veterinary Medicine, University of Illinois Urbana-Champaign, were enrolled in this study. Mares were housed either in the Horse Teaching Unit of the Department of Animal Sciences (*n* = 11) or the Veterinary Teaching Hospital of the University of Illinois Urbana-Champaign (*n* = 12). In the Horse Teaching Unit, mares were kept in stalls during the night and turned out into paddocks during the day. In the Veterinary Teaching Hospital, mares were kept in stalls and hand-walked in the clinic twice/day. All mares received a free choice of hay and water. All mares were bred with fresh extended semen by artificial insemination and had ovulation confirmed via ultrasonographic examinations performed every other day. 

### 2.2. Foaling Management

Each mare was monitored daily from arrival until the first week postpartum as part of the routine care. Mares were examined daily for signs of impending parturition, such as relaxation of the tail head and sacral ligaments, lengthening and softening of the vulva, engorged prominence of the mammary vein, and teat distension. Foaling was assisted, and any fetal mal-disposition was rapidly corrected. After parturition, each mare had the udder thoroughly cleaned with wetted cotton and dried. After that, colostrum was harvested to assess its quality using a colostrometer (Animal Reproduction Systems, Chino, CA, USA). Protruding portions of the fetal membranes were tied and recovered after being released from the mare. Immediately after release, fetal membranes from all mares were examined macroscopically for completeness and to rule out lesions (e.g., placentitis) that could have altered the electrolyte and pH profile in the subject mares [18]. Following the placental release, a quick vaginal palpation was performed to assess any trauma to the mare’s reproductive tract. Furthermore, the foals were watched closely to ensure proper nursing (nurse by 2 h after birth, one to two nurses per hour) then, each foal had blood samples collected to evaluate the transfer of passive immunity (IgG levels) from 12 h to 24 h after birth. 

### 2.3. Sampling and Determining the pH of Mammary Gland Secretions 

Immediately before harvesting the MGS, mares had udders washed with water-soaked cotton pieces. Each sampling of pre-foaling mammary gland secretions was carried out in a 50 mL conical tube as previously described [3,4,18]. Small aliquots (0.5 to 2 mL) of pre-foaling MGS were collected once a day (5 pm) until the pH dropped below 7.5; at this point, they were collected twice a day (am and pm). Depending on the amount of pre-foaling secretions yielded, one or two aliquots (ranging from 0.5 to 1.8 mL) were placed in 2 mL Eppendorf vials and preserved at −20 °C for further electrolyte evaluation. Determination of pH was achieved with the use of a portable pH meter (Compact pH METER B-71X, HORIBA Scientific, LAQUAtwin, Kyoto, Japan) coupled with a flat sensor (Sensor model S010, HORIBA Scientific) commonly used for research and clinical practice in our hospital [4,19]. Calibrations were performed immediately before each analysis using standard pH solutions according to the manufacturer’s instructions (pH 4.0 and pH 7.0 at 25 °C, HORIBA Scientific). Between analyses, the sampling well was thoroughly rinsed with bi-distilled water; excess water was drained by flipping the hand-held pH meter and then gently drying with a paper towel.

### 2.4. Analysis of MGS Electrolytes from Periparturient Mares

Retrospectively, samples collected seven days before and at the time of foaling were subjected to analyses of calcium (Ca^2+^), sodium (Na^+^), potassium (K^+^), and magnesium (Mg^2+^) using an automated Analyzer (AU 480 Beckman Coulter, CH 1260 Nyon, Switzerland). Our laboratory previously used this commercial analyzer to measure MGS [3]. These electrolytes were selected based on previous studies demonstrating the changes of these electrolytes in periparturient mares [3,8]. Samples were thawed at room temperature, filtered, and placed in the analyzer. 

### 2.5. Evaluation of Storage Temperature in the pH of MGS from Pre-Foaling Mares 

Aliquots containing 5–10 mL were harvested to determine the effects of temperature on the pH of MGS. Immediately after collection, samples were divided into three storage conditions: 37 °C incubator, 21 °C room temperature, and 5 °C (refrigerator). These three temperatures were chosen to emulate three possible field conditions that a practitioner working with a periparturient mare would face: a warm environment, a moderate environment, or a cold environment to transport samples to a site where pH can be measured. Ideally, each practitioner should carry a pH meter, but at times, neither method (pH strips or meter) is available, or the practitioner may have other priorities on a farm visit before being able to measure the pH. For the three storage temperatures, pH was measured (as described above) at 0, 15, 30, 45, and 60 min after storage and then repeated hourly for 10 h post-collection. A wide range of pH 8 to 6.5 was included in this evaluation to avoid bias towards one particular exact pH value. Thus, the chosen pH groups were as follows: 8 (range 7.8–8.2), 7.5 (range 7.3–7.7), 7 (6.7–7.2), and 6.5 (6.2–6.6)

### 2.6. Characterization of Milk pH during the First Week Postpartum in Mares 

Starting the day after parturition, all mares had MGS samples collected to evaluate pH as described above. Each aliquot was assessed within 15 min of the collection using the hand-held pH meter (Compact pH METER B-713X, HORIBA Scientific, LAQUAtwin, Kyoto, Japan).

### 2.7. Statistical Analyses

The data were normally distributed and analyzed using R (version 4.0.2). The pH was analyzed using mixed models for the seven days of pre-foaling. Fixed effects included storage conditions (three temperatures) and continuous effects included time (measured at multiple intervals). When significance was detected, post hoc comparisons were made with pairwise *t*-tests with Bonferroni Adjustment. The different models examined two-way interactions between fixed (storage conditions) and continuous effects (time). The significance was set as *p* ≤ 0.05. All the data were expressed as mean ± SEM. 

## 3. Results

All mares presented an average gestational length (mean 350 ± 3.5 days, range 334–368 days). Two mares had no records of the ovulation dates, but both mares had delivered normal foals uneventfully. All mares but two had uneventful parturitions; two minor dystocia (i.e., foot nape and a unilateral incomplete elbow extension) were immediately corrected, and the foals were delivered without complications. After a thorough examination, all placentas presented grossly normal and were confirmed to be intact. All foals had a regular passive immune transfer (>800 mg/dL) by 12 to 24 h postpartum. There was a reduction in pH leading to foaling (*p* < 0.05, Figure 1). The relative change was 13% from −7 d to 24 h pre-foaling. The overall pH on day 0 was 6.49 ± 0.2 (6.2–8.3) pH units; 85.7% (18/21) of the mares foaled with a pH ≤ 7. There was an increase in pH from foaling until 7 days postpartum (*p* < 0.001, Figure 2). There was an effect of time on pH from 15 min to 4 h compared to hour 0 for pH 8 (*p* < 0.05, Table 1). For pH 7.5, there was an effect of time from 15 min to 10 h in comparison to hour 0 (*p* < 0.05, Table 1 and Table 2). Additionally, pH 7 presented an effect of time from 3 to 10 h compared to hour 0 (*p* < 0.05, Table 1 and Table 2), while for pH 6.5, the effect of time was noted from 2 to 10 h in comparison to hour 0 (*p* < 0.05, Table 1 and Table 2). All pH levels demonstrated an effect of time from 5 to 10 h compared to hour 0 (*p* < 0.05, Table 2). Regarding storage temperature, pH 8 showed an effect at hours 2 to 4 at 37 °C (*p* < 0.05, Table 1), pH 7 from 2 to 10 h at 37 °C (*p* < 0.05, Table 2), and pH 6.5 from 2 to 10 h at both 37 °C and 22 °C (*p* < 0.05, Table 2). No effect of temperature was observed for pH 7.5. There was an effect of time on all the electrolytes (*p* < 0.05) (Figure 3 and Figure 4). There was a decrease in Na^+^ concentrations and an increase in K^+^, Mg^+^, and Ca^+^ concentrations 24 h pre-foaling (*p* < 0.05). Nineteen mares (88.9%) exhibited an inversion of Na^+^ and K^−^ 24 h before foaling. One mare foaled with a pH of 7, and two mares had pH levels above 7 (8.3 and 7.6). Notably, one mare with a high pH foaled without a Na^+^ and K^−^ inversion.

## 4. Discussion

This study was set forth to answer critical clinical questions; two are physiological, and the other is practical. In the first physiological question, we demonstrated that all mares had sodium potassium inversion and acidic pH at the time of foaling. In the second physiological question, we showed that milk pH is neutral in the first seven days postpartum; the clinical significance of this remains to be determined. In addition, with the final and practical question of this study, we demonstrated that the MGS can be measured with minimal variation for MGS stored at three different temperatures, except at the pH of ~7.5, which went up to ~8 as soon after the storage. This present study is the first to address these two physiological and practical questions about the pH of MGS in periparturient mares. 

Before this present study and another recent study by our group [16], it was unknown whether mares without sodium–potassium inversion before parturition would exhibit this inversion at the time of foaling, and whether the pH would be acidic at foaling for mares with an alkaline pH 6–12 h before foaling. We demonstrated herein and previously [3] that approximately 89% of mares had a sodium–potassium inversion 24 h before parturition. In contrast, the remaining mares did not exhibit a sodium–potassium inversion up to the parturition. The second physiological question answered in this present study was that mare milk is neutral starting 24 h postpartum and becomes slightly alkaline 5 days after foaling. At the same time, this does not confirm the acidic pH of the colostrum at the time of parturition, suggesting that this may play a role in the absorption of immunoglobulin by the neonate as previously suggested by another study [14]; this huge variation from ~6.5 to 7 could also reflect the difference in protein composition. 

The final practical question was whether MGS stored at various temperatures could later be assessed for pH, which might differ from the initial value at time 0. This component of the study used four arbitrarily defined pH ranges based on apparent clinical significance. For example, mares with MGS pH ~8 are not remotely close to foaling under physiological conditions. Mares with MGS pH ~7.5 typically will foal in 24–48 h or longer and mares with pH ~7 are on the verge of foaling in 24 h, whereas mares with pH ~6.5 are about to deliver [3,4,14,16]. In addition, we chose three storage temperatures to mimic what would be the best conditions to store samples: at 5 °C, mimicking a practitioner transporting a pH sample in an iced box; at 22 °C, mimicking someone living in a moderate climate and a temperature that most cars would be set for in a comfortable climate; and, finally, at 37 °C, mimicking a hot climate environment.

Interestingly, overall, the temperature and time seem to affect the pH of MGS for most ranges. The pH ~7.5 had a much more profound variation of 0.5 units up, bringing the mean values from ~7.5 at time zero to ~8 within minutes and remaining at this temperature for the remainder of the evaluation. However, the remaining pH ranges were stable during storage at either temperature. Fortunately, the pH of ~7.5 is not very common for mares to foal out, and pHs of 7.5 and 8 place the mare in a similar timeline for foaling; thus, if the ~7.5 is overestimated to be ~8, this would not affect the ability to use the pH of MGS to detect impending parturition. 

## 5. Conclusions

In conclusion, this study demonstrated that the majority of the mares had sodium potassium inversion and acidic pH at the time of foaling. Milk pH is neutral until four days postpartum, becoming slightly alkaline afterward; the clinical significance of this remains to be determined. The pH of MGS can be measured with variation stored at three different temperatures, except at the pH of ~7.5, which went up to ~8 as soon after the storage. The present study is the first to address these two physiological and practical questions about the pH of MGS, providing foundational data that paves the way for the clinical application of MGS; regardless of the effect of storage temperature over time.

## Figures and Tables

**Figure 1 animals-14-02598-f001:**
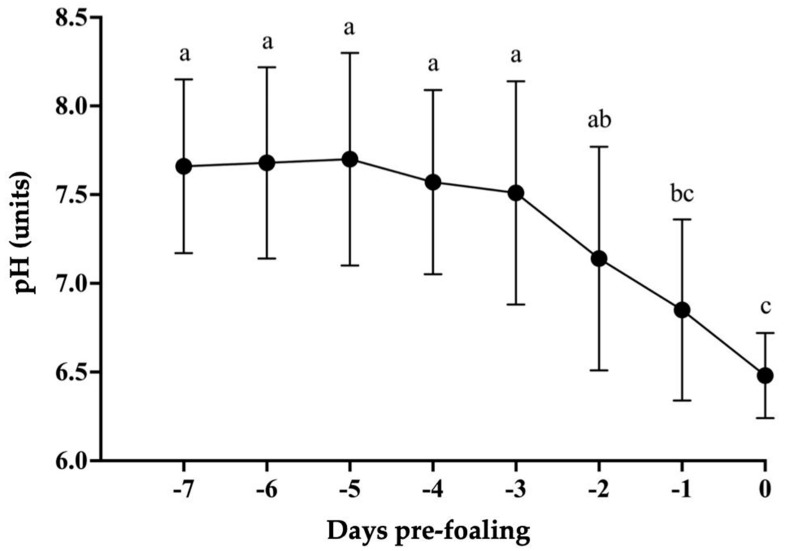
Measurements of pH seven days before and at the time of parturition in mares carrying and delivering normal foals. There was a reduction in pH leading to foaling (*p* < 0.001). Superscript letters (^a^, ^ab^, ^bc^, ^c^) denote statistical difference (*p* < 0.05).

**Figure 2 animals-14-02598-f002:**
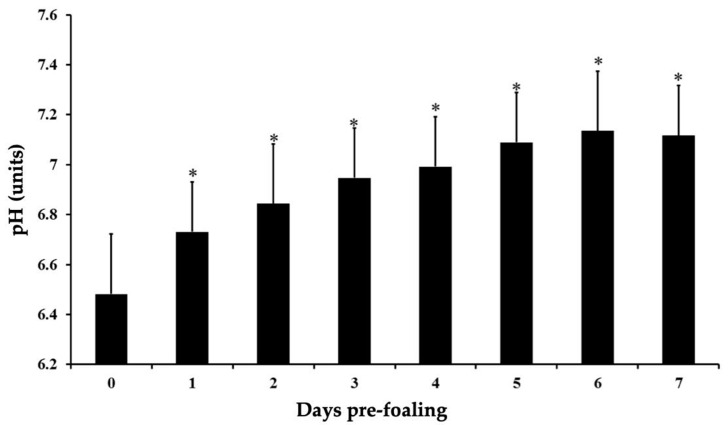
Measurements of pH at the time of parturition and during the first seven days postpartum in mares carrying and delivering normal foals. Asterisks (*) denote statistical difference (*p* < 0.05).

**Figure 3 animals-14-02598-f003:**
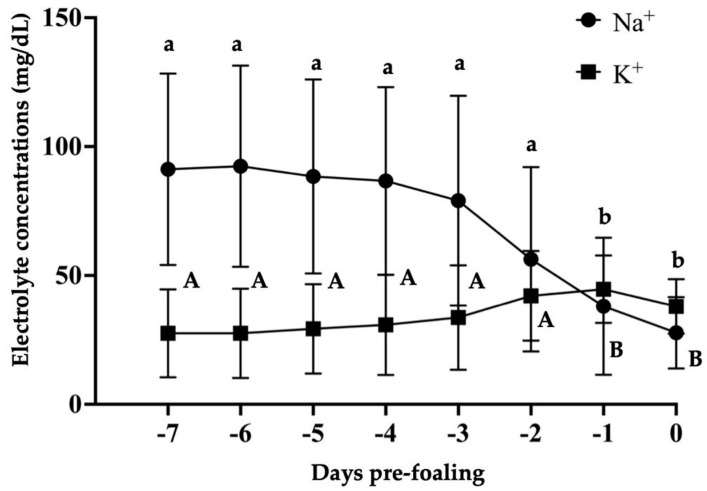
Sodium and potassium concentrations in the mammary gland secretions in mares from 7 days prepartum to the time of foaling. Different superscripts (^A^, ^a^, ^B^, ^b^) denote statistical significance (*p* < 0.05).

**Figure 4 animals-14-02598-f004:**
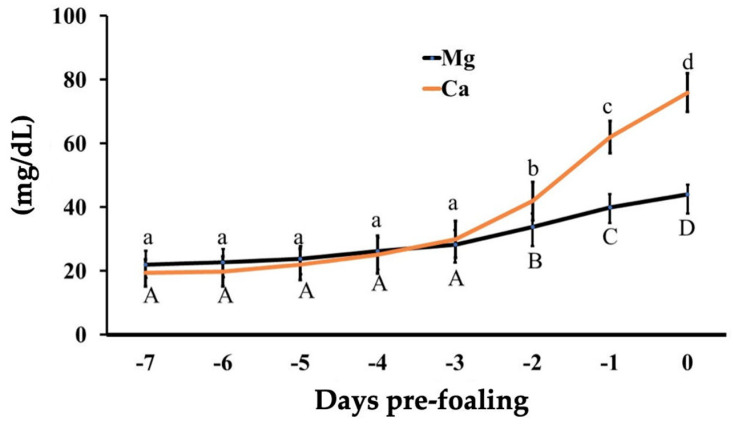
Calcium and magnesium concentrations in mammary gland secretions in mares from 7 days prepartum to the time of foaling. Superscript letters (^A^, ^a^, ^B^, ^b^, ^C^, ^c^, ^D^, ^d^) denote statistical significance (*p* < 0.05).

**Table 1 animals-14-02598-t001:** Mammary gland secretions were harvested from 21 periparturient mares. The ranges of pH included ~8 (range 7.8–8.2), ~7.5 (range 7.3–7.7), ~7 (6.7–7.2), and ~6.5 (6.2–6.6). The evaluations of each of the samples were performed at 0 min, 15 min, 30 min, 45 min, 60 min, and then hourly from 2 to 10 h post-harvesting. The analyses 5 to 10 h are depicted in Table 2.

pH	Temp	0 min	15 min	30 min	45 min	60 min	2 h	3 h	4 h
~8	5 °C	8 ± 0.1 ^a,x^	8.2 ± 0.1 ^b,x^	8.3 ± 0.1 ^b,x^	8.3 ± 0.1 ^b,x^	8.2 ± 0.1 ^b,x^	8.3 ± 0.1 ^b,x^	8.3 ± 0.1 ^b,x^	8.3 ± 0.1 ^b,x^
	22 °C	8 ± 0.1 ^a,x^	8.1 ± 0.1 ^a,x^	8.3 ± 0.1 ^b,x^	8.3 ± 0.1 ^b,x^	8.3 ± 0.1 ^b,x^	8.2 ± 0.1 ^b,x^	8.3 ± 0.1 ^b,x^	8.3 ± 0.11 ^b,x^
	37 °C	8 ± 0.1 ^a,x^	8.1 ± 0.1 ^a,x^	8.2 ± 0.1 ^a,x^	8.2 ± 0.1 ^a,x^	8.2 ± 0.1 ^b,x^	8. ± 0.1 ^b,y^	8.2 ± 0.1 ^b,y^	8.2 ± 0.11 ^b,y^
~7.5	5 °C	7.5 ± 0.1 ^a,x^	8 ± 0.1 ^b,x^	8 ± 0.1 ^b,x^	8 ± 0.1 ^b,x^	8 ± 0.1 ^b,x^	8.0 ± 0.1 ^b,x^	8.0 ± 0.1 ^b,x^	8.0 ± 0.1 ^b,x^
	22 °C	7.5 ± 0.1 ^a,x^	7.9 ± 0.2 ^b,x^	7.9 ± 0.2 ^b,x^	7.9 ± 0.1 ^b,x^	7.9 ± 0.1 ^b,x^	7.9 ± 0.1 ^b,x^	7.9 ± 0.12 ^b,x^	7.9 ± 0.12 ^b,x^
	37 °C	7.5 ± 0.1 ^a,x^	7.9 ± 0.1 ^b,x^	7.9 ± 0.1 ^b,x^	7.9 ± 0.1 ^b,x^	7.9 ± 0.1 ^b,x^	7.9 ± 0.1 ^b,x^	7.9 ± 0.1 ^b,x^	7.9 ± 0.1 ^b,x^
~7	5 °C	6.9 ± 0.1 ^a,x^	7 ± 0.07 ^a,x^	7 ± 0.1 ^a,x^	7 ± 0.1 ^a,x^	7 ± 0.1 ^a,x^	7 ± 0.1 ^a,x^	7.1 ± 0.1 ^b,x^	7.1 ± 0.1 ^b,x^
	22 °C	6.9 ± 0.1 ^a,x^	6.9 ± 0.1 ^a,x^	6.9 ± 0.1 ^a,x^	6.9 ± 0.1 ^a,x^	6.9 ± 0.1 ^a,x^	7 ± 0.1 ^a,x^	7.2 ± 0.1 ^b,x^	7.1 ± 0.1 ^b,x^
	37 °C	6.9 ± 0.1 ^a,x^	6.9 ± 0.1 ^a,x^	6.9 ± 0.1 ^a,x^	6.9 ± 0.1 ^a,x^	6.9 ± 0.1 ^a,x^	7 ± 0.1 ^a,y^	6.9 ± 0.1 ^b,y^	6.9 ± 0.1 ^b,y^
~6.5	5 °C	6.5 ± 0.1 ^a,x^	6.6 ± 0.1 ^a,x^	6.6 ± 0.1 ^a,x^	6.6 ± 0.1 ^a,x^	6.6 ± 0.1 ^a,x^	6.6 ± 0.1 ^b,x^	6.66 ± 0.04 ^b,x^	6.6 ± 0.1 ^b,x^
	37 °C	6.5 ± 0.1 ^a,x^	6.5 ± 0.1 ^a,x^	6.5 ± 0.1 ^a,x^	6.5 ± 0.1 ^a,x^	6.5 ± 0.1 ^a,x^	6.5 ± 0.1 ^a,y^	6.55 ± 0.04 ^a,y^	6.5 ± 0.1 ^a,y^
	22 °C	6.5 ± 0.1 ^a,x^	6.6 ± 0.1 ^a,x^	6.6 ± 0.1 ^a,x^	6.6 ± 0.1 ^a,x^	6.6 ± 0.1 ^a,x^	6.5 ± 0.1 ^a,y^	6.5 ± 0.03 ^a,y^	6.5 ± 0.1 ^a,y^

Different superscripts (^a^, ^b^) within the same row denote a statistical difference in comparison with the initial sample at time 0 min. Different superscripts (^x^, ^y^) within the same column denote significance between storage temperatures (*p* < 0.05).

**Table 2 animals-14-02598-t002:** Mammary gland secretions were harvested from 21 periparturient mares. The ranges of pH included ~8 (range 7.8–8.2), ~7.5 (range 7.3–7.7), ~7 (6.7–7.2), and ~6.5 (6.2–6.6). The evaluations of each sample were performed at 0 min, 15 min, 30 min, 45 min, and 60 min, and then hourly from 2 to 10 h post-harvesting. The analyses from 0 min to 4 h are depicted in Table 1.

pH	Temp	0 min	5 h	6 h	7 h	8 h	9 h	10 h
~8	5 °C	8.0 ± 0.1 ^a,x^	8.4 ± 0.1 ^b,x^	8.4 ± 0.1 ^b,x^	8.4 ± 0.1 ^b,x^	8.5 ± 0.1 ^b,x^	8.4 ± 0.1 ^b,x^	8.4 ± 0.1 ^b,x^
	22 °C	8.0 ± 0.1 ^a,x^	8.3 ± 0.1 ^b,x^	8.4 ± 0.1 ^b,x^	8.4 ± 0.1 ^b,x^	8.4 ± 0.1 ^b,x^	8.4 ± 0.1 ^b,x^	8.4 ± 0.1 ^b,x^
	37 °C	8 ± 0.1 ^a,x^	8.2 ± 0.1 ^b,y^	8.2 ± 0.1 ^b,y^	8.2 ± 0.13 ^b,y^	8.3 ± 0.12 ^b,y^	8.3 ± 0.1 ^b,y^	8.3 ± 0.1 ^b,y^
~7.5	5 °C	7.5 ± 0.1 ^a,x^	8.0 ± 0.1 ^b,x^	8.0 ± 0.1 ^b,x^	8.1 ± 0.1 ^b,x^	8.1 ± 0.1 ^b,x^	8.1 ± 0.1 ^b,x^	8.1 ± 0.1 ^b,x^
	22 °C	7.5 ± 0.1 ^a,x^	8.0 ± 0.12 ^b,x^	8.0 ± 0.1 ^b,x^	8.0 ± 0.1 ^b,x^	8.0 ± 0.1 ^b,x^	8.1 ± 0.1 ^b,x^	8.1 ± 0.1 ^b,x^
	37 °C	7.5 ± 0.1 ^a,x^	7.9 ± 0.1 ^b,x^	7.9 ± 0.1 ^b,x^	7.9 ± 0.1 ^b,x^	7.9 ± 0.1 ^b,x^	7.9 ± 0.1 ^b,x^	7.9 ± 0.1 ^b,x^
~7	5 °C	6.9 ± 0.1 ^a,x^	7.1 ± 0.1 ^b,x^	7.1 ± 0.1 ^b,x^	7.1 ± 0.1 ^b,x^	7.1 ± 0.1 ^b,x^	7.2 ± 0.1 ^b,x^	7.2 ± 0.1 ^b,x^
	22 °C	6.9 ± 0.1 ^a,x^	7.1 ± 0.1 ^b,x^	7.0 ± 0.1 ^b,x^	7.0 ± 0.1 ^b,x^	7.0 ± 0.1 ^b,x^	7.0 ± 0.1 ^b,x^	7.0 ± 0.1 ^b,x^
	37 °C	6.9 ± 0.1 ^a,x^	6.9 ± 0.1 ^b,y^	7 0 ± 0.1 ^b,y^	6.9 ± 0.1 ^b,y^	6.9 ± 0.1 ^b,y^	6.9 ± 0.1 ^b,y^	6.9 ± 0.1 ^b,y^
~6.5	5 °C	6.5 ± 0.1 ^a,x^	6.6 ± 0.03 ^b,x^	6.6± 0.1 ^b,x^	6.6 ± 0.1 ^b,x^	6.6 ± 0.1 ^b,x^	6.6 ± 0.1 ^b,x^	6.6 ± 0.1 ^b,x^
	22 °C	6.5 ± 0.1 ^a,x^	6.5 ± 0.1 ^a,y^	6.5 ± 0.1 ^a,y^	6.5 ± 0.1 ^a,y^	6.5 ± 0.1 ^a,y^	6.5 ± 0.1 ^a,y^	6.5 ± 0.1 ^b,y^
	37 °C	6.5 ± 0.1 ^a,x^	6.5 ± 0.1 ^a,y^	6.5 ± 0.1 ^a,y^	6.5 ± 0.1 ^a,y^	6.5 ± 0.1 ^a,y^	6.5 ± 0.1 ^a,y^	6.5 ± 0.1 ^a,y^

Different superscripts (^a^, ^b^) within the same row denote a statistical difference compared with the initial sample at 0 min. Different superscripts (^x^, ^y^) within the same column denote significance between temperature (*p* < 0.05).

## Data Availability

Data is unavailable due to privacy or ethical restrictions.

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
