# Peer review of "The Effect of Different Storage Temperatures over Time on the pH of Mammary Gland Secretions in Periparturient Mares"

_animals, 2024, doi:10.3390/ani14172598_

Round 1
Reviewer 1 Report
Comments and Suggestions for Authors
This is an interesting study summarizing and correlating milk pH and electrolyte concentrations from milk gland secretions with temperature during prepartum, at the time of parturition and in the first week postpartum. Developing a reliable and accurate method for determining the time of parturition in mares is still a challenge in veterinary sciences and practice as well in horse breeding. Thus, the objective of this manuscript is timely and important and the data have important meanings, both scientific and practical ones.
Although, the work is an extension of a previous studies by this other research groups, the data, may in concern summarize realistic and reliable measurements and methods to detect impending parturition in mares. Therefore, the assumptions and goals of this work are very important and present, not only of great cognitive but above all practical importance
The article is well written, easy to understand and contains important and recent knowledge. Materials and methods have been described detailed enough. This is a well-organized and experimentally well done study The experiments appear to have been carried out carefully and the methods generally sound. The data are well analyzed described and discussed. Relevant literature has been cited.
In general, the rationale, methodologies, and data are solid.
I have only few issues there should be addressed.
P1/L11 and P2/L70: This same abbreviation MGS has been used differently mammary gland electrolytes (MGS) or mammary gland secretions (MGS) Please correct it.
Conclusions: The conclusions are rather a summary of the results than an indication of the importance of the data and research for science and practice. Please complete this
Author Response
Open Review
(x) I would not like to sign my review report
( ) I would like to sign my review report
Quality of English Language
( ) I am not qualified to assess the quality of English in this paper
( ) English very difficult to understand/incomprehensible
( ) Extensive editing of English language required
( ) Moderate editing of English language required
( ) Minor editing of English language required
(x) English language fine. No issues detected
|
Yes |
Can be improved |
Must be improved |
Not applicable |
|
|
Does the introduction provide sufficient background and include all relevant references? |
(x) |
( ) |
( ) |
( ) |
|
Is the research design appropriate? |
(x) |
( ) |
( ) |
( ) |
|
Are the methods adequately described? |
(x) |
( ) |
( ) |
( ) |
|
Are the results clearly presented? |
(x) |
( ) |
( ) |
( ) |
|
Are the conclusions supported by the results? |
(x) |
( ) |
( ) |
( ) |
General comments and suggestions for authors
This is an interesting study summarizing and correlating milk pH and electrolyte concentrations from milk gland secretions with temperature during prepartum, at the time of parturition and in the first week postpartum. Developing a reliable and accurate method for determining the time of parturition in mares is still a challenge in veterinary sciences and practice as well in horse breeding. Thus, the objective of this manuscript is timely and important and the data have important meanings, both scientific and practical ones.
Although, the work is an extension of a previous studies by this other research groups, the data, may in concern summarize realistic and reliable measurements and methods to detect impending parturition in mares. Therefore, the assumptions and goals of this work are very important and present, not only of great cognitive but above all practical importance
The article is well written, easy to understand and contains important and recent knowledge. Materials and methods have been described detailed enough. This is a well-organized and experimentally well done study The experiments appear to have been carried out carefully and the methods generally sound. The data are well analyzed described and discussed. Relevant literature has been cited.
In general, the rationale, methodologies, and data are solid.
I have only few issues there should be addressed.
Reply: We thank the reviewer for the comments
Queries:
Query #1: P1/L11 and P2/L70: This same abbreviation MGS has been used differently mammary gland electrolytes (MGS) or mammary gland secretions (MGS) Please correct it.
Reply: We thank you for the comment, the typo error was addressed.
Query #2: Conclusions: The conclusions are rather a summary of the results than an indication of the importance of the data and research for science and practice. Please complete this
Reply: Thank you for your comment. We added a sentence at the conclusion enlightening the applicability of the technique over the “physiological variations”. Read as follows: “The present study is the first to address these two physiological and practical questions about the pH of MGS, providing foundational data that paves the way for the clinical application of MGS; regardless of the effect of storage temperature over time.

Reviewer 2 Report
Comments and Suggestions for Authors
This study evaluates the effect of pH and electrolyte changes overtime in mammary gland secretion in periparturient mares. This is not novel, and the study of how the pH changes overtime after MGS storage adds little value to the field, from the previous publications from the same group.
The statistics and Result section should be improved as suggested in the comments below.
Line 31: close bracket “)” after potassium determined
Line 171: “is not available”
Experimental design
More data should be given about the mares used in the study: inclusion criteria, at least range of weighs, ages. What was the reproductive history? If they were brought to the VTH, some might have had problems in previous foalings?
Statistics:
Were parametric or non-parametric tests used? No information about normality tests are given.
The analyses of pH over time should be performed with a repeated statement to account for autocorrelation between sequential observations (GLM), and report P values for effect of group (Temperature), Time and interactions.
To test difference between two consecutive days (Figure 2, etc). Paired t-test (normally distributed data) or Wilcoxon ranked test should be performed.
Results:
Line 190: The number of mares reported (n = 15 + 2) does not match with the number of mares described in the M&M (Line 113). What about the other 4 mares? Did they foal out of the ranges of gestation length or were they excluded from the study?
Line 197: Please give minimum and maximum for the pH at Day 0 from the group of mares. Did every mare foal with a pH below 7?
Data from Tables 1 and 2 would be clearer if presented in a Figure (scatter plot with a line and errors).
There is no reference to any Figure in the results section. There results about the MGS of postpartum mares is not reported in the text of the results section.
Discussion:
Line 237: These data are not reported in the Results section, only in Figure 2. Figure 2 indicate that the average milk pH on Day1 to 3 are below 7. What was the cut off to consider it neutral?
Lines 247-248: This data is not shown in the results section. From the Figures, it cannot be ascertain what percentage of mares foaled without sodium potassium inversion. Please give ranges and medians or specify in the text (results) what percentage of mares foaled with high pH and without inversion.
The lack of primiparous mares and the implication on the results should be discussed and acknowledged as limitation of the study.
If one of the main objectives of the study was to investigate the pH of milk after foaling, why was not correlated with the IgGs absorption as suggested in the introduction?
Author Response
Open Review
(x) I would not like to sign my review report
( ) I would like to sign my review report
Quality of English Language
( ) I am not qualified to assess the quality of English in this paper
( ) English very difficult to understand/incomprehensible
( ) Extensive editing of English language required
( ) Moderate editing of English language required
( ) Minor editing of English language required
(x) English language fine. No issues detected
|
Yes |
Can be improved |
Must be improved |
Not applicable |
|
|
Does the introduction provide sufficient background and include all relevant references? |
(x) |
( ) |
( ) |
( ) |
|
Is the research design appropriate? |
(x) |
( ) |
( ) |
( ) |
|
Are the methods adequately described? |
( ) |
( ) |
(x) |
( ) |
|
Are the results clearly presented? |
( ) |
( ) |
(x) |
( ) |
|
Are the conclusions supported by the results? |
(x) |
( ) |
( ) |
( ) |
Comments and Suggestions for Authors
General comments
This study evaluates the effect of pH and electrolyte changes overtime in mammary gland secretion in periparturient mares. This is not novel, and the study of how the pH changes overtime after MGS storage adds little value to the field, from the previous publications from the same group.
The statistics and Result section should be improved as suggested in the comments below.
Reply: We thank the reviewer for the comments and suggestions. However, this is the first study that evaluates the physiological variations of MGS over time and indirectly validates its use under specific temperature and time conditions.
Queries
Query #1: Line 31: close bracket “)” after potassium determined
Reply: The closing bracket was added as suggested
Query #2:Line 171: “is not available”
Reply: “not” was added as suggested.
Experimental design
Query #3. More data should be given about the mares used in the study: inclusion criteria, at least range of weighs, ages. What was the reproductive history? If they were brought to the VTH, some might have had problems in previous foalings?
Reply: We thank the reviewer for the comment. Unlike Texas, Illinois has fewer horse breeders who foal at home. Thus, owners opting to foal mares at the VTH is a conservative and safer choice, not necessarily due to problematic foaling history. All mares in the study were multiparous mares with no history of dystocia
Statistics:
Query #4. Were parametric or non-parametric tests used? No information about normality tests are given.
Reply: We thank you for the comment. A sentence indicating normality was added at the statistic session.
Query #5. The analyses of pH over time should be performed with a repeated statement to account for autocorrelation between sequential observations (GLM), and report P values for effect of group (Temperature), Time and interactions.
Reply: We appreciate the comment. The data was analysed with a mixed model accounting for the autocorrelation between sequential observations. A sentence was added in the results session clarifying the effects and interactions. Read as follows: “Overall, there was no effect of time, storage temperature, or association time: storage on the pH (P<0.05) (Table 1 and 2), except for pH ~7.5”
Query #6. To test the difference between two consecutive days (Figure 2, etc). Paired t-test (normally distributed data) or Wilcoxon ranked test should be performed.
Reply: Since there was no overall effect of storage temperature over time, we elected to use the pairwise t-test with Bonferroni adjustment (more conservative and controls for Type I error), ensuring that the observed difference is statistically robust (pH 7.5 between 0 and 15 min).
Results:
Query #7. Line 190: The number of mares reported (n = 15 + 2) does not match with the number of mares described in the M&M (Line 113). What about the other 4 mares? Did they foal out of the ranges of gestation length or were they excluded from the study?
Reply: Thank you. That was a typo and was removed from the text. Twenty-one mares foaled out.
Query #8. Line 197: Please give minimum and maximum for the pH at Day 0 from the group of mares. Did every mare foal with a pH below 7?
Reply: Thank you for the suggestion. The information was added and read as follows: “The overall pH on day 0 was 6.49± 0.2 (6.2 - 8.3) pH units, 85.7% (18/23) of the mares foaled with a slightly acidic pH below 7”
Query #9. Data from Tables 1 and 2 would be clearer if presented in a Figure (scatter plot with a line and errors).
Reply: We appreciate the comment however we elected to keep the result in tables.
Query #10. There is no reference to any Figure in the results section. There results about the MGS of postpartum mares is not reported in the text of the results section.
Reply. We thank the reviewer. The results session was rewritten and the references of the figures were included. Please read as follows: There was an effect of time on the pH from 7 days to foaling (p<0.05, fig 1). The relative change was 13 % from -7 d to 24 h pre-foaling. The overall pH on day 0 was 6.49± 0.2 (6.2 - 8.3) pH units; 85.7% (18/23) of the mares foaled with a slightly acidic pH below 7. There was an increase in pH from foaling until 7 days postpartum (P<0.001, fig 2). Overall, there was no effect of storage temperature, or association time: storage on the pH (P<0.05) (Table 1 and 2). There was an effect of time (0 min and 15 min) for pH ~7.5 (p<0.05). There was an effect of time on all the electrolytes (P<0.05) (Figs 3-4). There was a decrease in Na+ concentrations and an increase in K+, Mg+, and Ca+ concentrations 24 h pre-foaling (P<0.05).
Discussion:
Query #11. Line 237: These data are not reported in the Results section, only in Figure 2. Figure 2 indicate that the average milk pH on Day1 to 3 are below 7. What was the cut off to consider it neutral?
Reply: We thank the reviewer. However, the information is indeed in the text L210-212: There was an effect of time on all the electrolytes (P<0.05) (Figs 3-4). There was a decrease in Na+ concentrations and an increase in K+, Mg+, and Ca+ concentrations 24 h pre-foaling (P<0.05)”. From 7 to 7.5
Query #12. Lines 247-248: This data is not shown in the results section. From the Figures, it cannot be ascertained what percentage of mares foaled without sodium potassium inversion. Please give ranges and medians or specify in the text (results) what percentage of mares foaled with high pH and without inversion.
Reply: The authors thank the reviewers for the comments. A sentence was added clarifying the results please, read as follows: “Nineteen mares (88.9%) exhibited an inversion of Na+ and K-. One mare foaled with a pH of 7, and two mares had pH levels above 7 (8.3 and 7.6). Notably, one mare with a high pH foaled without a Na+ and K- inversion.”
Query #13 The lack of primiparous mares and the implication on the results should be discussed and acknowledged as limitation of the study.
Reply: We thank you for the comment however, the MGS pH values for primiparous mares and young multiparous are pretty similar. The age of the mare plays a more important role in the pH values than parity.
Query #14 If one of the main objectives of the study was to investigate the pH of milk after foaling, why was not correlated with the IgGs absorption as suggested in the introduction?
Reply: We thank the suggestion however the objectives of this study were to determine pH and electrolyte concentrations from MGS collected prepartum and at the time of parturition, to characterize mare milk pH in the first week postpartum, and, to evaluate pre-foaling MGS pH at three storage temperatures. We were not interested in any correlation between pH and IgG absorption in this particular study (a very limited number of mares, would be nearly impossible to find correlations/associations) since we have another manuscript under review focused on this topic.

Reviewer 3 Report
Comments and Suggestions for Authors
REVIEW
“The pH of mammary gland secretions in periparturient mares is acidic at the time of parturition, remains neutral during the first week postpartum, and does not vary with storage tempera-4 ture after harvesting”
BRIEF SUMMARY AND GENERAL COMMENTS:
The present study addresses changes in pH and electrolytes in the secretion of the mammary gland of mares during the peripartum period. Understanding these changes has the potential for practical applications in terms of greater survival of foals and productivity in this segment.
Knowledge in this area is relatively recent and allows for new knowledge or expansion/deepening in understanding biology and adjacent physiology.
However, the authors' current approach to the subject needs to present more details (as the data could), which could contribute to a more comprehensive understanding of the issue.
Also, some of the presented data need to be reviewed because they do not support some conclusions. The paper could be presented logically or sequentially, demanding less effort for the reader to understand the research. Similarly, technical terms are not standardized (e.g., Mamary gland secretions/ milk/ colostrum; periparturient/time of parturition).
The suggestion is to rewrite the material, emphasizing a more detailed and structured data presentation.
It would be important to have the work reviewed by an English-speaking proofreader.
SPECIFIC COMMENTS:
The presence of a space between the period and the beginning of a new sentence occurred several times. Please check the entire text regarding this.
Tables and Figures should be placed following the first time they appear in the text (for example, Tables 1 and 2 should be placed before the Figures). The titles of Tables and Figures need to be rewritten, following a scientific pattern. The layout and size of the Figures should be reviewed. For example, Figures 1 and 2 could present the same pattern, as well as Figures 3 and 4, because they present similar data, making it easier for the reader to compare information visually.
Title
The title is long. The presented data do not support the latest statement.
Abstract
The abstract could be clearer and could be more concise and organized. Some information presented in the abstract is not provided in the text
- Lines 22-26: The presented statements are presumptions instead of hypotheses.
Introduction
It would be important to present factors that could influence milk pH, like nutrition and infections.
- Line 71: Is the word “seminal” correct? Maybe the intended word was “several”?
Materials and methods
This topic could be improved by being presented in chronological order, which would make the text more straightforward for the reader.
It would be important to describe the age, improve the description of nutritional management (which hay, mineralization, without grains?), time of the year, bacteriological status of the teats (clinical/ subclinical mastitis)
Please describe with more detail the post-foaling management, especially regarding sample collections (was there a pattern regarding collection period, teats cleaning, separation of foals
- Line 149: The explanation is not relevant to the paper. The suggestion is to exclude it.
- Line 174: Please describe better the methodology. Was the ph considered as a class? Please inform the number of samples in each pH class. In the case of pH as a class, is the statistical methodology suitable?
Results
This topic could be explored more, presenting more detailed data on overall pH and relative changes. Also, it would be easier for the reader if the data were presented in the same order as the objectives.
Figures and Tables should be presented right after they are presented for the first time in the text.
The titles of tables and figures must follow a scientific style.
- Lines 196-197: The presented data did not support this statement.
Discussion
This topic should also be improved.
- Lines 235-236: Is this data described elsewhere in the text? Predefining which pH is considered acidic, neutral or alkaline would be necessary.
- Line 241: Please inform the origin of this data in the text.
- Line 242: Please insert some biography or biological/ physiological background to support this statement.
- Lines 244-247: Please rewrite to become clearer for the reader.
- Lines 248-249: Please rewrite to make it clearer for the reader.
- Lines 249-251: Please inform the origin of this data in the text.
- Lines 251-252: Please inform the origin of this data in the text.
- Lines 255-256: Please insert some biography or biological/ physiological background to support this statement.
- Lines 257-258: Please rewrite to become clearer for the reader.
- Lines 259-267: Please transfer these sentences to the Materials and Methods section.
- Lines 268-275: Please inform the origin of this data in the text. The presented Tables do not support these data.
Conclusions
The conclusion should not repeat the results. Instead, it should present applications of the presented data. Please rewrite it.
Comments on the Quality of English LanguageIt would be important to have the work reviewed by an English-speaking proofreader.
Author Response
REVIEW
“The pH of mammary gland secretions in periparturient mares is acidic at the time of parturition, remains neutral during the first week postpartum, and does not vary with storage tempera-4 ture after harvesting”
BRIEF SUMMARY AND GENERAL COMMENTS:
The present study addresses changes in pH and electrolytes in the secretion of the mammary gland of mares during the peripartum period. Understanding these changes has the potential for practical applications in terms of greater survival of foals and productivity in this segment.
Knowledge in this area is relatively recent and allows for new knowledge or expansion/deepening in understanding biology and adjacent physiology.
However, the authors' current approach to the subject needs to present more details (as the data could), which could contribute to a more comprehensive understanding of the issue.
Also, some of the presented data need to be reviewed because they do not support some conclusions. The paper could be presented logically or sequentially, demanding less effort for the reader to understand the research. Similarly, technical terms are not standardized (e.g., Mamary gland secretions/ milk/ colostrum; periparturient/time of parturition).
The suggestion is to rewrite the material, emphasizing a more detailed and structured data presentation.
It would be important to have the work reviewed by an English-speaking proofreader.
Reply: We thank the reviewer. We elected pertinent suggestions to address and ensure clarity to the manuscript.
SPECIFIC COMMENTS:
Query 1# The presence of a space between the period and the beginning of a new sentence occurred several times. Please check the entire text regarding this.
Reply: Thank you for the comment. Using double spacing enhances readability and is highly valued by European reviewers however, it was removed as suggested.
Query 2# Tables and Figures should be placed following the first time they appear in the text (for example, Tables 1 and 2 should be placed before the Figures). The titles of Tables and Figures need to be rewritten, following a scientific pattern. The layout and size of the Figures should be reviewed. For example, Figures 1 and 2 could present the same pattern, as well as Figures 3 and 4, because they present similar data, making it easier for the reader to compare information visually.
Reply: We thank the reviewer for the comment. The results session was rewritten under the suggestion.
Title
Query 3# The title is long. The presented data do not support the latest statement.
Reply: Since the overall effect of storage under temperature was not appreciated, we believe that the latest statement in the title is supported.
Abstract
Query 4# The abstract could be clearer and could be more concise and organized. Some information presented in the abstract is not provided in the text
- Lines 22-26: The presented statements are presumptions instead of hypotheses.
Reply: We specify the objectives and outline the hypothesis to ensure clarity
Introduction
Query 6# It would be important to present factors that could influence milk pH, like nutrition and infections.
- Line 71: Is the word “seminal” correct? Maybe the intended word was “several”?
Reply. The authors thank for the suggestions. However, there was no scientific evidence-based to support the effect of diet on the MGS pH pre-foaling; little is known about cows (dry matter intake and milk pH). Furthermore, the unhealthy udder directly affects the pH of the MGS, since, the study considered only healthy mares we do not agree with adding this to the intro. The typo was corrected (L 143).
Materials and methods
Query 8# This topic could be improved by being presented in chronological order, which would make the text more straightforward for the reader.
Reply: We appreciated the comment however we believe that the session was well organized to ensure clarity.,
Query 9# It would be important to describe the age, improve the description of nutritional management (which hay, mineralization, without grains?), time of the year, bacteriological status of the teats (clinical/ subclinical mastitis)
Reply: The age of the mare was included; the nutritional management was already in the text. Bacteriological tests were not performed and the udder/teats were evaluated clinically only, with no mares presenting clinical signs of inflammation/infection. We don’t agree that the time of the year(season) would have any influence and was not accounted for in the study design since the mares were all kept indoors at night times with controlled temperature and humidity. We also do not pursue the knowledge to judge how the climate would directly or indirectly influence the pH of MGS in mares.
Query 10# Please describe with more detail the post-foaling management, especially regarding sample collections (was there a pattern regarding collection period, teats cleaning, separation of foals
Reply: We believe that this information is already included in the text with sufficient evidence to ensure comprehension.
- Query 11# Line 149: The explanation is not relevant to the paper. The suggestion is to exclude it.
Reply. The explanation ensures clarity and highlights the usability of the device in a clinical trial.
- Query 12# Line 174: Please describe better the methodology. Was the ph considered as a class? Please inform the number of samples in each pH class. In the case of pH as a class, is the statistical methodology suitable?
Reply: Thank you for the comment. We believe that considering pH as a class (discrete pH groups) in the mixed model analysis is statistically suitable and provides a clear framework for understanding the effects of different pH levels, time points, and storage conditions on the pH stability of MGS This approach ensures that the study accurately captures the underlying biological and temporal variations.
Results
Query 13# This topic could be explored more, presenting more detailed data on overall pH and relative changes. Also, it would be easier for the reader if the data were presented in the same order as the objectives.
Figures and Tables should be presented right after they are presented for the first time in the text.
- Query 14# Lines 196-197: The presented data did not support this statement.
Reply. We believe it does support since there was not an overall effect but a single effect of time on the pH 7.5 (overall effect vs simple effects). We believe that the overall mixed model might indicate no significant effect or interaction due to the averaging of effects across all groups and time points. Furthermore, the statement “Overall, there was no effect of storage temperature, or association time: storage on the pH (P<0.05) (Table 1 and 2). There was an effect of time (0 min and 15 min) for pH ~7.5 (p<0.05)” is suitable to explain the overall vs. single effects.
Discussion
This topic should also be improved.
- Query 15# Lines 235-236: Is this data described elsewhere in the text? Predefining which pH is considered acidic, neutral or alkaline would be necessary.
Reply:
- Query 16# Line 241: Please inform the origin of this data in the text.
Reply: It is in tables 1 and 2. Displaying the little variation on pH 7.5 between 0 and 15 min (8 pH units) and remaining stable over time.
- Query 17# Line 242: Please insert some biography or biological/ physiological background to support this statement.
- Query 18# Lines 244-247: Please rewrite to become clearer for the reader.
Reply. We thank you for the comment and improved clarity. Please read as follows: “Before the present study and another recent study by our group [15], it was unknown whether mares without sodium-potassium inversion before parturition would exhibit this inversion at the time of foaling, and whether the pH would be acidic at foaling for mares with an alkaline pH 6-12 hours before foaling. We demonstrated herein and previously [3] that approximately 80% of mares had a sodium-potassium inversion before parturition.”
- Query 19# Lines 248-249: Please rewrite to make it clearer for the reader.
Reply: read as follow: “In contrast, the remaining mares did not exhibit a sodium-potassium inversion up to the last sampling before parturition. However, all mares showed a sodium-potassium inversion and an acidic pH at the time of foaling.”
- Query 20# Lines 249-251: Please inform the origin of this data in the text.
Reply: The sentence was rewritten and the results supporting the statement were added to the result session. Please read as follows: We demonstrated herein and previously [3] that approximately 89% of mares had a sodium-potassium inversion before parturition. In contrast, the remaining mares did not exhibit a sodium-potassium inversion up to the parturition.
- Query 21# Lines 251-252: Please inform the origin of this data in the text.
Reply: This was included in the objectives in the introduction.
- Query 22# Lines 255-256: Please insert some biography or biological/ physiological background to support this statement.
Reply: The reference is in the text. Korouse, K., Murase, H., Sato, F., Ishimaru, M., Kotoyori, Y., Tsujimura, K. & Nambo, Y. (2013) Comparison of pH and refractometry index with calcium concentrations in preparturient mammary gland secretions of mares. Journal of the American Veterinary Medical Association 242, 242-248.
- Query 23# Lines 257-258: Please rewrite to become clearer for the reader.
Reply: Please read as follows: “The final practical question was whether MGS stored at various temperatures could later be assessed for pH, which might differ from the initial value at time 0. This component of the study used four arbitrarily defined pH ranges based on apparent clinical significance”
- Query 24# Lines 259-267: Please transfer these sentences to the Materials and Methods section.
Reply: We elected to leave it as it is since the sentence is directly connected with the next paragraph ensuring clarity and good readability.
- Query 25# Lines 268-275: Please inform the origin of this data in the text. The presented Tables do not support these data.
Reply: Tables 1 and 2 support the results, also well described in the results session. “There was an effect of time on the pH from 7 days to foaling (p<0.05, fig 1). The relative change was 13 % from -7 d to 24 h pre-foaling. The overall pH on day 0 was 6.49± 0.2 (6.2 - 8.3) pH units; 85.7% (18/21) of the mares foaled with a pH £ 7. There was an increase in pH from foaling until 7 days postpartum (P<0.001, fig 2). Overall, there was no effect of storage temperature, or association time: storage on the pH (P<0.05) (Table 1 and 2). There was an effect of time (0 min and 15 min) for pH ~7.5 (p<0.05)”
Conclusions
Query 26# The conclusion should not repeat the results. Instead, it should present applications of the presented data. Please rewrite it.
Reply: The conclusion was rewritten. Please read as follows: “In conclusion, this study demonstrated that all mares had sodium potassium inversion and acidic pH at the time of foaling. Milk pH is neutral in the first seven days postpartum; the clinical significance of this remains to be determined. The pH of MGS can be measured with minimal variation stored at three different temperatures, except at the pH of ~7.5, which went up to ~8 as soon after the storage; however, the clinical significance of this variation can likely be negligible. The present study is the first to address these two physiological and practical questions about the pH of MGS, providing foundational data that paves the way for the clinical application of MGS; regardless of the effect of storage temperature over time”
Comments on the Quality of English Language
Query 27# It would be important to have the work reviewed by an English-speaking proofreader.

Round 2
Reviewer 3 Report
Comments and Suggestions for Authors
GENERAL
Thank you very much for the time and effort in checking and correcting the pointed questions. Unfortunately, some points were not sufficiently addressed, and they will be presented in a more detailed manner.
The focus will be on addressing queries that need attention.
Query 2# Tables and Figures should be placed following the first time they appear in the text (for example, Tables 1 and 2 should be placed before the Figures). The titles of Tables and Figures need to be rewritten, following a scientific pattern. The layout and size of the Figures should be reviewed. For example, Figures 1 and 2 could present the same pattern and Figures 3 and 4 because they present similar data, making it easier for the reader to compare information visually.
Reply: We thank the reviewer for the comment. The results session was rewritten under the suggestion.
Revisor reply: The titles of Tables and Figures should be presented in scientific format. They can explain what is shown. The interpretation of the graphics and tables should be presented only in the text.
TITLE
Query 3# The title is long. The presented data do not support the latest statement.
Reply: Since the overall effect of storage under temperature was not appreciated, we believe that the latest statement in the title is supported.
Revisor reply: The data presented in Tables 1 and 2 do not permit us to conclude that "The pH of Mammary Gland Secretions in Periparturient Mares … Does Not Change with Storage Temperature over Time" once in pH class 8, there was a variation in pH after 2 hours (at three temperatures); in pH class 7, there was variation after 3 hours (at three temperatures); and in pH class 6.5 there was a variation after 2 hours at a temperature of 5ºC and after 10 hours at a temperature of 22ºC (P<0.05). The only pH class that did not vary (P>0.05) was the pH 6.5 class stored at 37ºC until 10 hours after collection.
This research did not evaluate the extent to which the magnitude of this variation is relevant in biological terms. Therefore, speculation in the title or conclusions is not correct. However, it is a pertinent topic in the body of the discussion, especially if it is based on other research already carried out.
INTRODUCTION
Query 6# It would be important to present factors that could influence milk pH, like nutrition and infections.
Reply. The authors thank for the suggestions. However, there was no scientific evidence-based to support the effect of diet on the MGS pH pre-foaling; little is known about cows (dry matter intake and milk pH). Furthermore, the unhealthy udder directly affects the pH of the MGS, since, the study considered only healthy mares we do not agree with adding this to the intro.
Revisor reply: Studies on equid colostrum/milk refer to the importance of diet in the chemical composition of colostrum/milk, which can indirectly influence its pH. In addition, they also indicate the importance of calving order and environmental conditions in the chemical composition of milk/colostrum, which in turn have a potential impact on its pH.
https://doi.org/10.1111/j.1740-0929.2011.00930.x
https://doi.org/10.1016/j.cveq.2021.01.001
https://doi.org/10.1371/journal.pone.0238921
https://doi.org/10.1016/j.idairyj.2020.104781
https://doi.org/10.1007/978-3-031-35271-3_1
https://doi.org/10.1016/0958-6946(94)00008-D
MATERIALS AND METHODS
Query 9# It would be important to describe … bacteriological status of the teats (clinical/ subclinical mastitis)
Reply: Bacteriological tests were not performed and the udder/teats were evaluated clinically only, with no mares presenting clinical signs of inflammation/infection.
Revisor reply: It is documented that subclinical mastitis in mares and cows can impact pH and electric conductivity.
https://doi.org/10.3390/ani12040440
https://doi.org/10.3390/molecules27238631
Query 10# Please describe with more detail the post-foaling management, especially regarding sample collections (was there a pattern regarding the collection period, teats cleaning, separation of foals
Reply: We believe this information is already included in the text with sufficient evidence to ensure comprehension.
Revisor reply: The provided description, "From 12 to 24 hours postpartum, each foal had blood samples collected to evaluate the transfer of passive immunity", does not favour the replication of this study. For example, Reference #13 describes it in a more detailed manner:
"Preparturient mammary gland secretion sampling—The preparturient mammary gland secretion sample was collected for ten days prior to foaling. A minimum sample quantity of 2 mL was obtained daily, between 6:00 AM and 7:00 AM, until the pH of the preparturient mammary gland secretion samples decreased to < 7.0. Subsequently, the preparturient mammary gland secretion samples were obtained twice daily, between 6:00 AM and 7:00 AM and again between 3:00 PM and 4 PM, until parturition. Some mares did not produce sufficient quantities of pre-parturient mammary gland secretion ten days before foaling, so samples were obtained when the amount of mammary gland secretions became sufficient."
https://doi.org/10.2460/javma.242.2.242
RESULTS
Query 13# The titles of Tables and Figures need to be rewritten, following a scientific pattern.
Revisor reply: The titles of Tables and Figures should be presented in scientific format. They can explain what is shown in it. The interpretation of the graphics and tables should be presented only in the text.
Table 1 does not present statistics for the pH 8 class at 15 min of storage. This needs to be corrected.
Figures 3 and 4 still lack statistical markers (*); see Lines 212-217 of the corrected version.
Query 14# Lines 196-197: The presented data did not support this statement.
Reply. We believe it does support since there was not an overall effect but a single effect of time on the pH 7.5 (overall effect vs simple effects). We believe that the overall mixed model might indicate no significant effect or interaction due to the averaging of effects across all groups and time points. Furthermore, the statement "Overall, there was no effect of storage temperature, or association time: storage on the pH (P<0.05) (Table 1 and 2). There was an effect of time (0 min and 15 min) for pH ~7.5 (p<0.05)" is suitable to explain the overall vs. single effects.
Revisor reply: The statement "Overall, there was no effect of storage temperature, or association time: storage on the pH (p < 0.05) (Table 1 and 2)" (lines 197-198 of corrected version)" is not accurate to describe de presented pH variation, regarding storage temperature nor pH variation over time. Please correct the sentence.
The data presented in Tables 1 and 2 shows that pH was affected (P < 0.05) by storage temperature and over time in all considered pH classes, except for class 6,5 at 37ºC, regarding time and pH class 7.5 considering temperature.
For instance, in the analysis of Tables 1 and 2, considering storage temperature, there was no difference during the first hour (P>0.05). After two h of storage, there was a difference in pH classes 8, 7 and 6,5 (P<0.05); this trend repeats after 3h, 4h, 5h, 6h, 7h, 9h and 10h. This statistic shows that overall, after the first hour, temperature impacted the pH measure (even if the magnitude of the variation was small).
Considering time effect over pH, the data presented in Tables 1 and 2 shows that in pH class 8, there was a variation (P<0.05) in pH after 2 hours (at three temperatures); in pH class 7, there was variation (P<0.05) after 3 hours (at three temperatures); and in pH class 6.5 there was a variation (P<0.05) after 2 hours at a temperature of 5ºC and after 10 hours at a temperature of 22ºC (P<0.05). The only pH class that did not vary (P>0.05) was the pH 6.5 class stored at 37ºC until 10 hours after collection.
Interestingly, the variation was described in the text as showing effect (P<0.05) instead of greater than 0.05 (line 198; corrected version).
New Query I# Lines 200-201 of corrected version: Figures 3 and 4 need to address the described mineral variation.
New Query II# Lines 193-194 of corrected version: The sentence "There was an effect of time on the pH from 7 days to foaling (p < 0.05, Figure 1)" is not accurate. Figure 1 illustrates the effect of time on the pH only from one day of pre-foaling. Please correct the sentence.
New Query III# Line 202 of corrected version: "Nineteen mares (88.9%) exhibited an inversion of Na+ and K− in the time of foaling" is inaccurate.
Figure 3 illustrates a Na-K inversion 24h pre-foaling that was maintained until foaling. It could seem like an irrelevant semantic matter, but one day can make all the difference in this case.
The sentence should be rewritten: "Nineteen mares (88.9%) exhibited an inversion of Na+ and K− 24 h pre-foaling."
DISCUSSION
Query 15# Lines 235-236: Is this data described elsewhere in the text?
Reply:
Revisor reply_a: The statement "we demonstrated that all mares had sodium potassium inversion … at the time of foaling."; (lines 240-241 of corrected version)" it is not accurate to describe the sodium and potassium variation at the time of foaling (11,1% of the mares did not present this inversion). Please correct the sentence.
Revisor reply_b: The statement "Nineteen mares (88.9%) exhibited an inversion of Na+ and K− in the time of foaling; (lines 240-241 corrected version)" does not seem to be accurate. Figure 3 illustrates a Na-K inversion 24h pre-foaling that was maintained until foaling. Please correct the sentence.
Revisor reply_c: The statement "we demonstrated that all mares had… and acidic pH at the time of foaling."; (lines 240-241 corrected version)" is not accurate in describing the sodium and potassium variation at the time of foaling, once this was not observed in 14,3% of the mares (line 196 of corrected version). Please correct the sentence.
Revisor reply_d: The statement "milk pH is neutral in the first seven days postpartum" (lines 240-241 corrected version) does not accurately describe the sodium and potassium variation at the time of foaling.
As presented in Figure 2, although the pH rose daily from the first day postpartum, it was neutral only on the fourth day and became alkaline from the fifth day onward. Please correct the sentence.
Query 21# Lines 251-252: Please inform the origin of this data in the text.
Reply: This was included in the objectives in the introduction
Revisor reply: According to the presented data, although the pH rises daily postpartum, the mare's milk becomes neutral only on the fourth day postpartum and turns alkaline from the fifth to the seventh day, instead of becoming neutral starting 24 h postpartum and up to seven days. Please correct it.
Query 25# Lines 268-275: Please inform the origin of this data in the text. The presented Tables do not support these data.
Reply: Tables 1 and 2 support the results, also well described in the results session. "There was an effect of time on the pH from 7 days to foaling (p<0.05, fig 1). The relative change was 13 % from -7 d to 24 h pre-foaling. The overall pH on day 0 was 6.49± 0.2 (6.2 - 8.3) pH units; 85.7% (18/21) of the mares foaled with a pH £ 7. There was an increase in pH from foaling until 7 days postpartum (P<0.001, fig 2). Overall, there was no effect of storage temperature, or association time: storage on the pH (P<0.05) (Table 1 and 2). There was an effect of time (0 min and 15 min) for pH ~7.5 (p<0.05)"
Revisor reply: Please refer to Revisor's reply to Query 14#.
CONCLUSIONS
New Query IV# Lines 281-282 of corrected version: Please correct the sentence "…all mares had sodium potassium inversion and acidic pH at the time of foaling" as stated in Revisor's reply to Querys 15#_a and c.
New Query V# Line 282 of corrected version: Please correct the sentence "Milk pH is neutral in the first seven days postpartum", as stated in Revisor's reply to Query 15#_d.
New Query VI# Lines 283-284 of corrected version: Please correct the sentence "The pH of MGS can be measured with minimal variation stored at three different temperatures", as stated in Revisor's reply to Query 14#.
New Query VII# Lines 285-286 of corrected version: It is well known that subtle pH changes can significantly impact physiological, clinical, and microbiological terms. This study did not focus on evaluating the impact of pH variations on clinical or laboratory parameters. Thus, this issue can be raised in the discussion, preferably based on research already carried out on that subject or presenting the question as a hypothesis. However, this statement does not fit in the conclusion of the article.
Please remove the sentence "the clinical significance of this variation can likely be negligible; line 285 of the corrected version.
Comments on the Quality of English LanguageMinor editing of English language
Author Response
Open Review
(x) I would not like to sign my review report
( ) I would like to sign my review report
Quality of English Language
( ) I am not qualified to assess the quality of English in this paper.
( ) The English is very difficult to understand/incomprehensible.
( ) Extensive editing of English language required.
( ) Moderate editing of English language required.
(x) Minor editing of English language required.
( ) English language fine. No issues detected.
|
Yes |
Can be improved |
Must be improved |
Not applicable |
|
|
Does the introduction provide sufficient background and include all relevant references? |
( ) |
(x) |
( ) |
( ) |
|
Is the research design appropriate? |
( ) |
(x) |
( ) |
( ) |
|
Are the methods adequately described? |
( ) |
(x) |
( ) |
( ) |
|
Are the results clearly presented? |
( ) |
( ) |
(x) |
( ) |
|
Are the conclusions supported by the results? |
( ) |
( ) |
(x) |
( ) |
Comments and Suggestions for Authors
GENERAL
Thank you very much for the time and effort in checking and correcting the pointed questions. Unfortunately, some points were not sufficiently addressed, and they will be presented in a more detailed manner.
The focus will be on addressing queries that need attention.
Query 2# Tables and Figures should be placed following the first time they appear in the text (for example, Tables 1 and 2 should be placed before the Figures). The titles of Tables and Figures need to be rewritten, following a scientific pattern. The layout and size of the Figures should be reviewed. For example, Figures 1 and 2 could present the same pattern and Figures 3 and 4 because they present similar data, making it easier for the reader to compare information visually.
Reply: We thank the reviewer for the comment. The results session was rewritten under the suggestion.
Revisor reply: The titles of Tables and Figures should be presented in scientific format. They can explain what is shown. The interpretation of the graphics and tables should be presented only in the text.
Reply: The titles were rewritten containing under a scientific format as suggested.
TITLE
Query 3# The title is long. The presented data do not support the latest statement.
Reply: Since the overall effect of storage under temperature was not appreciated, we believe that the latest statement in the title is supported.
Revisor reply: The data presented in Tables 1 and 2 do not permit us to conclude that "The pH of Mammary Gland Secretions in Periparturient Mares … Does Not Change with Storage Temperature over Time" once in pH class 8, there was a variation in pH after 2 hours (at three temperatures); in pH class 7, there was variation after 3 hours (at three temperatures); and in pH class 6.5 there was a variation after 2 hours at a temperature of 5ºC and after 10 hours at a temperature of 22ºC (P<0.05). The only pH class that did not vary (P>0.05) was the pH 6.5 class stored at 37ºC until 10 hours after collection.
This research did not evaluate the extent to which the magnitude of this variation is relevant in biological terms. Therefore, speculation in the title or conclusions is not correct. However, it is a pertinent topic in the body of the discussion, especially if it is based on other research already carried out.
Reply: The title was altered: “The Effect of Different Storage Temperatures on the pH of Mammary Gland Secretions in Periparturient Mares Over Time”
INTRODUCTION
Query 6# It would be important to present factors that could influence milk pH, like nutrition and infections.
Reply. The authors thank for the suggestions. However, there was no scientific evidence-based to support the effect of diet on the MGS pH pre-foaling; little is known about cows (dry matter intake and milk pH). Furthermore, the unhealthy udder directly affects the pH of the MGS, since, the study considered only healthy mares we do not agree with adding this to the intro.
Revisor reply: Studies on equid colostrum/milk refer to the importance of diet in the chemical composition of colostrum/milk, which can indirectly influence its pH. In addition, they also indicate the importance of calving order and environmental conditions in the chemical composition of milk/colostrum, which in turn have a potential impact on its pH.
https://doi.org/10.1111/j.1740-0929.2011.00930.x
https://doi.org/10.1016/j.cveq.2021.01.001
https://doi.org/10.1371/journal.pone.0238921
https://doi.org/10.1016/j.idairyj.2020.104781
https://doi.org/10.1007/978-3-031-35271-3_1
https://doi.org/10.1016/0958-6946(94)00008-D
Reply: A sentence was added.
MATERIALS AND METHODS
Query 9# It would be important to describe … bacteriological status of the teats (clinical/ subclinical mastitis)
Reply: Bacteriological tests were not performed and the udder/teats were evaluated clinically only, with no mares presenting clinical signs of inflammation/infection.
Revisor reply: It is documented that subclinical mastitis in mares and cows can impact pH and electric conductivity.
https://doi.org/10.3390/ani12040440
https://doi.org/10.3390/molecules27238631
Reply: It’s highly controversial how to diagnose subclinical mastitis in mares since factors can lead to the presence of PMN in the colostrum/milk (i.e., vaccine, hormones -weaning, colostrogenese). Plus, most of the studies are focused on milk and not on COLOSTRUM, those substance differs substantially in composition and cellularity. The study cited by you reported an initial increase right after birth and a slight increase during the weaning process again since it wasn’t evaluated directly it is IMPOSSIBLE to infer that it was subclinical mastitis; the literature under subclinical mastitis (diagnostics…) is very scarce and not established yet, especially because it’s quite rare in the horse due to anatomically and genetically features. In other domestic species, the increase in leukocytes right after birth is a part of the transfer of immunity. An increase in leukocytes which will be passed via colostrum to the offspring acting as “memory cells” colonizing different tissues making part of the host defense, and acting synergistically with IgG. The cellular component of colostrum is as important as the IgG in the TPI. It’s named the colostrogenese and begins several weeks before parturition and abruptly concludes near the time of parturition. This is described in bovine. Thus, it cannot be related to subclinical mastitis (transient increase in SCC) thus, it won’t be added to this manuscript.
Query 10# Please describe with more detail the post-foaling management, especially regarding sample collections (was there a pattern regarding the collection period, teats cleaning, separation of foals
Reply: We believe this information is already included in the text with sufficient evidence to ensure comprehension.
Revisor reply: The provided description, "From 12 to 24 hours postpartum, each foal had blood samples collected to evaluate the transfer of passive immunity", does not favour the replication of this study. For example, Reference #13 describes it in a more detailed manner:
"Preparturient mammary gland secretion sampling—The preparturient mammary gland secretion sample was collected for ten days prior to foaling. A minimum sample quantity of 2 mL was obtained daily, between 6:00 AM and 7:00 AM, until the pH of the preparturient mammary gland secretion samples decreased to < 7.0. Subsequently, the preparturient mammary gland secretion samples were obtained twice daily, between 6:00 AM and 7:00 AM and again between 3:00 PM and 4 PM, until parturition. Some mares did not produce sufficient quantities of pre-parturient mammary gland secretion ten days before foaling, so samples were obtained when the amount of mammary gland secretions became sufficient."
https://doi.org/10.2460/javma.242.2.242
Reply: We believe that the session provides sufficient evidence for repeatability. We did include a statement to ensure clarity regarding the foal management. Read as follows “Furthermore, the foals were watched closely to ensure proper nursing (nurse by 2 hrs after birth, one to two nurses per hour) then, each foal had blood samples collected to evaluate the transfer of passive immunity (IgG levels) from 12hr to 24 hrs after birth”
RESULTS
Query 13# The titles of Tables and Figures need to be rewritten, following a scientific pattern.
Revisor reply: The titles of Tables and Figures should be presented in scientific format. They can explain what is shown in it. The interpretation of the graphics and tables should be presented only in the text.
Table 1 does not present statistics for the pH 8 class at 15 min of storage. This needs to be corrected.
Figures 3 and 4 still lack statistical markers (*); see Lines 212-217 of the corrected version.
Reply: The markers were added for both figures and tables
Query 14# Lines 196-197: The presented data did not support this statement.
Reply. We believe it does support since there was not an overall effect but a single effect of time on the pH 7.5 (overall effect vs simple effects). We believe that the overall mixed model might indicate no significant effect or interaction due to the averaging of effects across all groups and time points. Furthermore, the statement "Overall, there was no effect of storage temperature, or association time: storage on the pH (P<0.05) (Table 1 and 2). There was an effect of time (0 min and 15 min) for pH ~7.5 (p<0.05)" is suitable to explain the overall vs. single effects.
Revisor reply: The statement "Overall, there was no effect of storage temperature, or association time: storage on the pH (p < 0.05) (Table 1 and 2)" (lines 197-198 of corrected version)" is not accurate to describe de presented pH variation, regarding storage temperature nor pH variation over time. Please correct the sentence.
The data presented in Tables 1 and 2 shows that pH was affected (P < 0.05) by storage temperature and over time in all considered pH classes, except for class 6,5 at 37ºC, regarding time and pH class 7.5 considering temperature.
For instance, in the analysis of Tables 1 and 2, considering storage temperature, there was no difference during the first hour (P>0.05). After two h of storage, there was a difference in pH classes 8, 7 and 6,5 (P<0.05); this trend repeats after 3h, 4h, 5h, 6h, 7h, 9h and 10h. This statistic shows that overall, after the first hour, temperature impacted the pH measure (even if the magnitude of the variation was small).
Considering time effect over pH, the data presented in Tables 1 and 2 shows that in pH class 8, there was a variation (P<0.05) in pH after 2 hours (at three temperatures); in pH class 7, there was variation (P<0.05) after 3 hours (at three temperatures); and in pH class 6.5 there was a variation (P<0.05) after 2 hours at a temperature of 5ºC and after 10 hours at a temperature of 22ºC (P<0.05). The only pH class that did not vary (P>0.05) was the pH 6.5 class stored at 37ºC until 10 hours after collection.
Interestingly, the variation was described in the text as showing effect (P<0.05) instead of greater than 0.05 (line 198; corrected version).
Reply: The results session was rewritten as suggested please, read as follow: “There was an effect of time on pH from 15 minutes to 4 hours in comparison to hour 0 for pH 8 (p < 0.05, Table 1). For pH 7.5, there was an effect of time from 15 minutes to 10 hours in comparison to hour 0 (p < 0.05, Table 1 and 2). Additionally, pH 7 presented an effect of time from 3 to 10 hours compared to hour 0 (p < 0.05, Table 1and 2), while for pH 6.5, the effect of time was noted from 2 to 10 hours in comparison to hour 0 (p < 0.05, Table 1 and 2). All pH levels demonstrated an effect of time from 5 to 10 hours when compared to hour 0 (p < 0.05, Table 2). Regarding storage temperature, pH 8 showed an effect at hours 2 to 4 at 37°C (p < 0.05, Table 1), pH 7 from 2 to 10 hours at 37°C (p < 0.05, Table 2), while pH 6.5 from 2 to 10 hours at both 37°C and 22°C (p < 0.05, Table 2). No effect of temperature was observed for pH 7.5”
New Query I# Lines 200-201 of corrected version: Figures 3 and 4 need to address the described mineral variation.
Reply: The superscript letters were added to clarify the figures.
New Query II# Lines 193-194 of corrected version: The sentence "There was an effect of time on the pH from 7 days to foaling (p < 0.05, Figure 1)" is not accurate. Figure 1 illustrates the effect of time on the pH only from one day of pre-foaling. Please correct the sentence.
Reply: Corrected as suggested now read as follow: “There was a reduction in pH, leading to foaling (p < 0.05, Figure 1).
New Query III# Line 202 of corrected version: "Nineteen mares (88.9%) exhibited an inversion of Na+ and K− in the time of foaling" is inaccurate.
Figure 3 illustrates a Na-K inversion 24h pre-foaling that was maintained until foaling. It could seem like an irrelevant semantic matter, but one day can make all the difference in this case.
The sentence should be rewritten: "Nineteen mares (88.9%) exhibited an inversion of Na+ and K− 24 h pre-foaling."
Reply: Thank you for the comment. The sentence was rewritten as suggested.
Query 15# Lines 235-236: Is this data described elsewhere in the text?
Revisor reply_a: The statement "we demonstrated that all mares had sodium potassium inversion … at the time of foaling."; (lines 240-241 of corrected version)" it is not accurate to describe the sodium and potassium variation at the time of foaling (11,1% of the mares did not present this inversion). Please correct the sentence.
Reply: Please read as follows in lines 299-305: We demonstrated herein and previously [3] that approximately 89% of mares had a sodium-potassium inversion before parturition. In contrast, the remaining mares did not exhibit a sodium-potassium inversion up to the parturition.
Revisor reply_b: The statement "Nineteen mares (88.9%) exhibited an inversion of Na+ and K− in the time of foaling; (lines 240-241 corrected version)" does not seem to be accurate. Figure 3 illustrates a Na-K inversion 24h pre-foaling that was maintained until foaling. Please correct the sentence.
Reply: Read as follows: “Approximately 89% of mares had a sodium-potassium inversion 24 hours before parturition”
Revisor reply_c: The statement "we demonstrated that all mares had… and acidic pH at the time of foaling."; (lines 240-241 corrected version)" is not accurate in describing the sodium and potassium variation at the time of foaling, once this was not observed in 14,3% of the mares (line 196 of corrected version). Please correct the sentence.
Reply: Corrected as suggested.
Revisor reply_d: The statement "milk pH is neutral in the first seven days postpartum" (lines 240-241 corrected version) does not accurately describe the sodium and potassium variation at the time of foaling.
As presented in Figure 2, although the pH rose daily from the first day postpartum, it was neutral only on the fourth day and became alkaline from the fifth day onward. Please correct the sentence.
Reply: Read as follows: “becomes neutral starting 24 h postpartum and getting slightly alkaline from 5 days after foaling”
Query 21# Lines 251-252: Please inform the origin of this data in the text.
Reply: This was included in the objectives in the introduction
Revisor reply: According to the presented data, although the pH rises daily postpartum, the mare's milk becomes neutral only on the fourth day postpartum and turns alkaline from the fifth to the seventh day, instead of becoming neutral starting 24 h postpartum and up to seven days. Please correct it.
Reply: same as reply d : Read as follows: “becomes neutral starting 24 h postpartum and getting slightly alkaline from 5 days after foaling”
Query 25# Lines 268-275: Please inform the origin of this data in the text. The presented Tables do not support these data.
Reply: Tables 1 and 2 support the results, also well described in the results session. "There was an effect of time on the pH from 7 days to foaling (p<0.05, fig 1). The relative change was 13 % from -7 d to 24 h pre-foaling. The overall pH on day 0 was 6.49± 0.2 (6.2 - 8.3) pH units; 85.7% (18/21) of the mares foaled with a pH £ 7. There was an increase in pH from foaling until 7 days postpartum (P<0.001, fig 2). Overall, there was no effect of storage temperature, or association time: storage on the pH (P<0.05) (Table 1 and 2). There was an effect of time (0 min and 15 min) for pH ~7.5 (p<0.05)"
Revisor reply: Please refer to Revisor's reply to Query 14#.
Reply: Addressed already in Query 14#.
CONCLUSIONS
New Query IV# Lines 281-282 of corrected version: Please correct the sentence "…all mares had sodium potassium inversion and acidic pH at the time of foaling" as stated in Revisor's reply to Querys 15#_a and c.
Reply: Already addressed in Query 15#.
New Query V# Line 282 of corrected version: Please correct the sentence "Milk pH is neutral in the first seven days postpartum", as stated in Revisor's reply to Query 15#_d.
Reply: Addressed in Query 15#
New Query VI# Lines 283-284 of corrected version: Please correct the sentence "The pH of MGS can be measured with minimal variation stored at three different temperatures", as stated in Revisor's reply to Query 14#.
Reply: Modified based on the query 14#.
New Query VII# Lines 285-286 of corrected version: It is well known that subtle pH changes can significantly impact physiological, clinical, and microbiological terms. This study did not focus on evaluating the impact of pH variations on clinical or laboratory parameters. Thus, this issue can be raised in the discussion, preferably based on research already carried out on that subject or presenting the question as a hypothesis. However, this statement does not fit in the conclusion of the article.
Please remove the sentence "the clinical significance of this variation can likely be negligible; line 285 of the corrected version.
Reply: Removed as suggested
Comments on the Quality of English Language
Minor editing of English language
Submission Date
06 June 2024
Date of this review
02 Aug 2024 19:43:43